# Monthly and seasonal rainfall (1963-2017) for a typhoon-influenced area in a Chinese karst basin

Chongxun Mo<sup>1, 2, 3</sup>, Yuli Ruan<sup>1, 2, 3,\*</sup>, Jiaqi He<sup>1, 2, 3</sup>, Guikai Sun<sup>1, 2, 3</sup>, Juliang Jin<sup>4</sup>

<sup>1</sup> College of architecture and civil engineering Guangxi University, Nanning, China

<sup>2</sup> Key Laboratory of Disaster Prevention and structural Safety of Ministry of Education, Nanning, China

<sup>3</sup> Guangxi Key Laboratories of Disaster Prevention and Engineering Safety, Nanning, China

<sup>4</sup> School of Civil Engineering, Hefei University of Technology, Hefei 230009, China

Correspondence to: Yuli Ruan (yuliruan777@163.com)

Abstract. Under the dual influence of global warming and typhoon weather, the characteristics of monthly and seasonal

- rainfall become more complicated and extreme, leading to more frequent rainfall and flood disasters. The objective of this study is to propose a framework for analysing the characteristics of rainfall and its correlation with typhoons. The proposed framework consists of an analysis of inner-annual distribution, inter-annual variation as well as correlation between rainfall and typhoons. Especially, a DVM method is proposed to compute the rationality and reliability of the abrupt change time analysis. Finally, the proposed framework was successfully implemented using a 55-year time series (1963-2017) of rainfall
- data recorded by 12 rain gauges in Chengbi River Basin (South China). The results are as follows. (1) The relative variability of monthly rainfall is relatively large (above 30%), and the rainfall tends to be centralized, and is mainly concentrated in July. (2) The monthly average rainfall is relatively stable, the seasonal rainfall decreases in spring and summer, while increases in autumn and winter. The abrupt change occurred during 1980s-1990s. The main periods of rainfall in summer and winter are shorter than those in spring and autumn. (3) The typhoon-caused rain is the main factor affecting the summer and autumn
- rainfall while the number of typhoons has the least influence. It is suggested that the impact of typhoon should be taken into consideration in the modelling of spatial and temporal evolution of hazardous rainfall evens and their social hazard assessment for a typhoon-influenced area .Besides, preventive measures should be strengthened for flood and waterlogging disasters and that the reservoir should be operated at different FLWLs during different flood sub seasons in Chengbi River Basin.

# 25 1 Introduction

The analysis of hydro meteorological elements has been a popular and frontier issue in recent years. Rainfall is an important factor affecting natural disasters, such as floods and droughts. With global climate change and the intensification of human activities, the characteristics of rainfall are more complex, and they have attracted the attentions of scholars all over the world. Many studies have shown that a study of rainfall characteristics mainly starts from two aspects: inner-annual

distribution and inter-annual variation.

Several indices can be applied to study the inner-annual distribution of rainfall, such as precipitation concentration degree (PCD), precipitation concentration period (PCP) and non-uniformity index. For example, the non-uniformity index and regulation coefficient were applied to analyse the annual distribution characteristics of rainfall in Henan province, China (Wang et al., 2007). Additionally, PCD, PCP and inhomogeneous distribution coefficient were analysed, and the annual

- distribution characteristics of rainfall were studied. The results showed that the distributions of heavy rain, rainstorms and erosive rainfall events were uneven (Gao et al., 2017). Many studies have shown that trend, abrupt change time and period can correctly represent the inter-annual variation characteristics of rainfall to some degree. Recently, some methods have been suggested for examining the trends of annual, seasonal and monthly rainfall (Luković et al., 2014). Additionally, trends of daily rainfall characteristics, including intensity, duration and frequency, were analysed in Finland (Irannezhad et al.,
- 2016). In Japan, researchers were interested both in the trends of rainfall and their relationship to moisture variations and topographic convergence (Iwasaki, 2015). For China, the trends in precipitation can vary from time to time and place to place (e.g.Zhang et al., 2007; Gao et al., 2012; Zeng et al., 2013; Wu et al., 2016). In addition, the abrupt change time (e.g.Tarhule and Woo, 2015; Zhang et al., 2009; Guerreiro et al., 2014; Peng et al., 2016) and period (Yu et al., 2012; Iqbal and Ali , 2013) of rainfall are also the focus of contemporary studies.
- The rainfall mechanism in Guangxi Zhuang Autonomous Region (Guangxi) are :(1) cyclonic (typhoon) activities, such as typhoon rain during the summer and autumn;(2) terrain blocking, such as orographic rain;(3) convective motion, such as thunderstorm in summer afternoon;(4) frontal activities, such as mould rains at the turn of spring and summer. Cyclonic activities and topographic barriers are the core mechanism of rainfall change in Guangxi. This study mainly focuses on typhoon. On one hand, Guangxi is located in the low latitudes of China; the rainfall is influenced by geographical location
- and monsoon circulation, especially typhoon, which frequently causes drought and waterlogging disasters. Therefore, some studies have focused on the characteristics of tropical cyclones (typhoons) and their influence. In 2005, regular data from 1960 to 1999 was used to study the intensity variation, maintaining time and disappeared-location characteristics of tropical cyclones for Guangxi region (Chen et al., 2005). Additionally, some researchers were interested in the statistical characteristics of typhoon-heavy rainfall in Beibu Gulf Area of Guangxi (He et al., 2007). In 2014, the CMA best track data
- from 1953 to 2012 were used to study the spatial-temporal distribution and inter-annual variation in tropical cyclones, and the results showed that frequency and maximum intensity of the TCs were decreasing, while the average intensity was increasing (Zhang et al.,2014). From these studies, we know that from 1960 to 2010, there were 60 occurrences of rainstorms with large ranges in this region, which brought serious economic losses and posed a threat to the life, health and safety of the local inhabitants. Additionally, Guangxi is located in tropical monsoon region, which is influenced by the alternation of
- winter and summer monsoons and a complex geographical environment, so severe drought disasters occur frequently. For example, there were 3 large drought disasters in Guangxi from 1998 to 2009 (Dong, 2014). This is also a proof that the intensity and ratio of extreme events caused by global climate change have increased. On the other hand, most of the basins in Guangxi are karst basins with shallow soil layers and steep hills. Natural disasters, such as soil erosion and landslides, are

closely related to rainfall on monthly and seasonal scales. Therefore, it is necessary and urgent to study the characteristics of monthly and seasonal rainfall.

In summary, rainfall is an important factor that leads to occurrence of floods, droughts and other natural disasters, especially under the background of global warming. Frequency of extreme weather events will likely continue to increase, so an

- analysis of the characteristics of rainfall is vital to take appropriate technical means to prevent the disasters caused by extreme rainfall. Rainfall is affected by geographical location and monsoon circulation, especially typhoon in Guangxi, and therefore characteristics of rainfall will be more complex than those in other places. However, there are few studies about the characteristics of rainfall and its relationship to typhoons in Guangxi karst basin at present. In this study, we will try to answer two questions: first, how are the characteristics of monthly and seasonal rainfall in the context of climate change?
- And second, whether the changes of rainfall have some relationship with typhoons or how typhoons affect the rainfall? The objectives are as follows: (1) analyse the inter-annual distribution of rainfall with the help of relative variability, PCD and PCP indices; (2) study the inter-annual variation characteristics (trends, abrupt change time and period) of rainfall by using Mann-Kendall method, hydrologic variation diagnosis system, and wavelet analysis; and (3) analyse the relationship between rainfall and typhoons using Pearson correlation coefficients and Grey relational analysis. The innovations of the
- study are: (1) it is the first study on monthly and seasonal rainfall characteristics and their correlations with typhoon of a karst-typhoon doubly affected area in southwest China ;(2) in addition, in the analysis of abrupt change time, we put forward a DVM method. This method can provide a more reasonable and reliable analysis of the initial determined abrupt change time, making the study more accurate and credible.

### 2 Study area and data

- Chengbi River Basin belongs to Xijiang River system, and the river originates from Qinglong Mountain in the northern area of Lingyun County, Baise City, Guangxi. The total area of this basin is 2087 km<sup>2</sup>, the average elevation is 650 m, the shape of the basin is similar to a rectangle, and the terrain is high in the northwest and low to the southeast. The basin covers typical karst limestone landforms, including an underground river, a falling water cave, and hilly landforms, high mountains and high vegetation coverage. The basin is located in subtropical monsoon climate zone, and therefore the climate is mild
- and the rainfall is abundant. The annual rainfall distribution is uneven, and the average annual rainfall reaches 1560 mm. The climate is abnormal, with gales and hail weather in April and high temperature, heavy rain and strong wind weather patterns from July to September, which very easily form disastrous floods. In this study, 12 monitoring stations are used: Ba Shou (BS) station, Bai Lian (BL) station, Xia Tang (XT) station, Lin He (LH) station, Ping Tang (PT) station, Hao Kun (HK) station, Nong Tang (NT) station, Chao Li (CL) station, Xia Jia (XJ) station, Lin Yun (LY) station, Dong He (DH) station and
- Jie Fu (JF) station (Figure 1). The 55-year (1963-2017) monthly and season rainfall data from each station are provided by the ChengBi River Reservoir administration, and the typhoon data (the number of typhoons per year, the typhoon maintaining time and the maximum typhoon wind speed) can be download from http://typhoon.nmc.cn. Then, following

5

methods are adopted to interpolate and extend the monitoring station with data missing: (1) The mean value of the adjacent data have been used to interpolate the monitoring stations with less data missing .(2) For the monitoring stations with more data missing, firstly, Ba Shou station has been used as reference station then use simple linear regression model (R2>0.95) to interpolate the data. The rationality and feasibility of the above methods have been verified by (Zhang et al., 2015). Finally the mean areal rainfall calculated by Thiessen polygon method is applied to this study.

Fig. 1 The location of Chengbi River Basin and the distribution of the rainfall stations, and the representative rainfall stations are shown as red stars

#### 3 Methods

#### 10 **3.1 Methods for inner-annual distribution analysis**

The inner-annual distribution characteristics of rainfall mainly include rainfall stability and concentration. The characteristics of rainfall stability can be reflected by its average relative variability, where the greater the average relative variation in rainfall, the more unstable the rainfall is (Conrad,1941; Sun,1986).

PCD and PCP are two important indexes that represent the inner-annual distribution of rainfall. The rainfall of each month is 15 taken as a vector, length of the vector is the size of the rainfall, while direction of the vector is the time of the rainfall. From January to December, the azimuth angle of each month is  $\theta = 0^\circ$ ,  $\theta = 30^\circ$ ,  $\theta = 60^\circ$ ,  $\dots \theta = 330^\circ$ , and the monthly