# Peer review of "Monthly and seasonal rainfall (1963-2017) for a typhoon-influenced area in a Chinese karst basin"

_Natural Hazards and Earth System Sciences, 2018_

## Referee Comment (RC1) · Anonymous Referee #1 · 29 Sep 2018

The present manuscript presents the characteristics of monthly and seasonal rainfall and its correlation with typhoons in a Chinese karst basin. The work is interesting and inspiring to the field of the understanding of natural hazards and their consequences. Over all, the applied methods are valid, the results and conclusions are presented in a clear and well-structured way. Therefore I recommend this paper to be publicated. And it is better if the authors consider the following mentioned remarks and further improve the manuscript before submitting the final version. The authors should provide at least some information about how typhoons affect the studied area in the introduction section. Besides, it could be useful for the reader if the authors present a flow chart of the proposed procedure.

[Figure]

2018-194, 2018.

---

## Referee Comment (RC2) · Anonymous Referee #2 · 8 Oct 2018

This study analyzed the characteristics of monthly and seasonal rainfall and their correlations with typhoon in a karst basin in southwest China. Over all, there are some drawbacks in the manuscript. It needs much more revisions for improvement of this manuscript. The comments are as follows: 1.The manuscript tries to answer two questions: first, how are the characteristics of monthly and seasonal rainfall in the context of climate change? second, whether the changes of rainfall have some relationship with typhoons or how typhoons affect the rainfall? However,the analyzed contents are not related to the climate change,and the rainfall changes with typhoons are not analyzed in detail. 2. What is "DVM method"? What is new for this method? It is not clearly indicated. 3."Many studies have shown that a study of rainfall characteristics mainly starts from two aspects: inner-annual distribution and inter-annual variation." This statement

is not correct. 4."mould rains" is not correct. 5.Page 3 Line 24ïijŽ"The basin is located in subtropical monsoon climate zone"is not consistent with "Guangxi is located in tropical monsoon region"in Page 2 Line 29.

---

## Author Comment (AC1) · 8 Nov 2018

Dear referee: Thank you very much for your help in processing the review of our manuscript (Manuscript ID nhess-2018-194). We have carefully read the thoughtful comments from you and found that these suggestions are helpful for us to improve our manuscript. The itemized responses to the comments are listed below. We hope that all these corrections and revisions would be satisfactory. Thanks a lot, again.

Responses to comments 1 The authors should provide at least some information about how typhoons affect the studied area in the introduction section. Reply: Thank you for your constructive and helpful suggestion. Since there is no specific introduction to the studied area in the introduction section.We provide the information about how typhoons

affect the studied area in the study area and data section.We hope this modification meets your requirements.(The revision are shown in blue.)

2 Besides, it could be useful for the reader if the authors present a flow chart of the proposed procedure. Reply: Thank you for your constructive and helpful suggestion. A flow chart of the proposed procedure has been provided.(figure 2) Sincerely,

Yuli Ruan

College of architecture and civil engineering Guangxi University Nanning, Guangxi, China, 530004

Please also note the supplement to this comment:
https://www.nat-hazards-earth-syst-sci-discuss.net/nhess-2018-194/nhess-2018-194-AC1-supplement.zip

---

## Author Comment (AC2) · 8 Nov 2018

Dear referee: Thank you very much for your help in processing the review of our manuscript (Manuscript ID nhess-2018-194).We have carefully read the thoughtful comments from you and found that these suggestions are helpful for us to improve our manuscript. The itemized responses to the comments are listed below. We hope that all these corrections and revisions would be satisfactory. Thanks a lot, again.

Responses to comments &1 The manuscript tries to answer two questions: first, how are the characteristics of monthly and seasonal rainfall in the context of climate change? Second, whether the changes of rainfall have some relationship with ty-phoons or how typhoons affect the rainfall? However, the analyzed contents are not

related to the climate change, and the rainfall changes with typhoons are not analyzed in detail. Reply: Thank you for your constructive and helpful suggestion. It should be noted that this study mainly focus on two issues, one is the characteristics of monthly and seasonal rainfall,another is the correlation between rainfall and typhoon.In order to avoid misunderstanding,the expressions about "Climate change and human activity" have been removed.We think that the study on the characteristics of rainfall change under the influence of climate change and human activities is an interesting subject, but it is also a very complicated one. It requires a lot of detailed meteorological data, such as temperature, evaporation, solar radiation and wind speed.This paper is not dedicated to studying issues related to climate change, but if the data is met we will take into account the impacts of climate change and human activities in future studies. In addition, we have supplemented the analysis of rainfall characteristics under the influence of typhoon according to your suggestions. Please refer to section 4.2.4 for details.(All the revised sentence are in green.)

& 2 What is"DVM method"? What is new for this method? It is not clearly indicated. Reply: Thank you for your constructive and helpful suggestion. The Degree of variability method (DVM method) is a proposed method that can analyse the rationality and reliability of the determined abrupt change time in this study as stated in the section 3.3.2,line 5.This method assumes that the rainfall should be very different before and after the abrupt change time,and is expressed by the degree of variability.The rationality and scientificity of this method lie in the multi-indexes,which can comprehensive reflect the variability of the rainfall before and after the abrupt change time. Concretely, 10 indexes of rainfall data, including EX (mathematical expectation), CV (deviation coefficient), CS (skewness coefficient), R10P (10% quantile rainfall), R70P (70% quantile rainfall), R90P (90% quantile rainfall), CDD (the longest continuous rain free days), CWD (the longest continuous rain days), DR20 (days of heavy rain (more than 20 mm/d per year on average)), and R20 (amount of heavy rain (more than 20 mm/d per year on average)) were used to evaluate the change degree of rainfall before and after the determined abrupt change time. The greater the variability degree of rainfall

before and after the abrupt change time, the more rational and reliable the time was determined to be.

& 3 "Many studies have shown that a study of rainfall characteristics mainly starts from two aspects: inner-annual distribution and inter-annual variation."This statement is not correct. Reply: Thank you for your constructive and helpful suggestion. The original sentence was corrected to "Many studies have focused on inner-annual distribution and inter-annual variation when analysed the rainfall characteristics."(the revised sentence are in yellow page1,line 28-29)

& 4 "mould rains" is not correct.. Reply: Thank you for your constructive and helpful suggestion. After investigation, it is generally believed that the mould rains mainly occur in the middle and lower reaches of the Yangtze river,China (Excluding guangxi), Taiwan, south-central Japan and South Korea.Therefore, we have modified the content of Guangxi's rainfall mechanism- deleted the content related to mould rains.

&5 "The basin is located in subtropical monsoon climate zone"is not consistent with"Guangxi is located in tropical monsoon region"in Page 2 Line 29. Reply: Thank you for your constructive and helpful suggestion. The sentence"Guangxi is located in tropical monsoon region"has be corrected to "Guangxi is located in subtropical mon- soon region."(the revised sentence are in yellow page 2,line 28-29)

Sincerely,

Yuli Ruan

College of architecture and civil engineering Guangxi University Nanning, Guangxi, China, 530004

Please also note the supplement to this comment:
https://www.nat-hazards-earth-syst-sci-discuss.net/nhess-2018-194/nhess-2018-194-AC2-supplement.zip